# Cadmium Body Burden and Inflammatory Arthritis: A Pilot Study in Patients from Lower Silesia, Poland

**DOI:** 10.3390/ijerph19053099

**Published:** 2022-03-06

**Authors:** Iwona Markiewicz-Górka, Małgorzata Chowaniec, Helena Martynowicz, Anna Wojakowska, Aleksandra Jaremków, Grzegorz Mazur, Piotr Wiland, Krystyna Pawlas, Rafał Poręba, Paweł Gać

**Affiliations:** 1Division of Environmental Health and Occupational Medicine, Department of Population Health, Wroclaw Medical University, 7 Mikulicza-Radeckiego St., 50-345 Wroclaw, Poland; malgorzata.chowaniec@onet.pl (M.C.); aleksandra.jaremkow@umw.edu.pl (A.J.); krystyna.pawlas@umw.edu.pl (K.P.); pawel.gac@umw.edu.pl (P.G.); 2Department of Rheumatology and Internal Medicine, Wroclaw Medical University, 213 Borowska St., 50-556 Wroclaw, Poland; piotr.wiland@umw.edu.pl; 3Department of Internal and Occupational Diseases, Hypertension and Clinical Oncology, Wroclaw Medical University, 213 Borowska St., 50-556 Wroclaw, Poland; helena.martynowicz@umw.edu.pl (H.M.); anna.wojakowska@umw.edu.pl (A.W.); grzegorz.mazur@umw.edu.pl (G.M.); rafal.poreba@umw.edu.pl (R.P.)

**Keywords:** cadmium, inflammatory arthritis, interleukin 10, rheumatoid arthritis, ankylosing spondylitis, psoriatic arthritis, cyclooxygenase-2, erythrocyte sedimentation rate, high-sensitivity C-reactive protein, 8-hydroxy-2′-deoxyguanosine

## Abstract

The purpose of this study was to determine the relationship between cadmium exposure and the likelihood of developing or exacerbating symptoms of inflammatory arthritis (IA). The study included 51 IA patients and 46 control subjects. Demographic and lifestyle data were collected. Haematological and biochemical parameters and blood cadmium levels (Cd-B) were determined. Cd-B correlated positively with age, smoking, living in a high-traffic area, and serum levels of inflammatory markers and negatively with mean corpuscular haemoglobin concentration (MCHC). The binary logistic regression model implied that high Cd-B (≥0.65 μg/L) is linked with an increased risk of IA in the studied population (odds ratio: 4.4). High levels of DNA oxidative damage marker (8-hydroxy-2’-deoxyguanosine) (≥7.66 ng/mL) and cyclooxygenase-2 (≥22.9 ng/mL) and frequent consumption of offal was also associated with increased risk of IA. High Cd-B was related to increased risk of disease symptoms onset in the group of IA patients, decreased the level of interleukin 10, and positively correlated with the disease activity. Increased Cd-B is associated with intensified inflammatory processes and decreased haemoglobin levels; in IA patients with decreased anti-inflammatory interleukin 10. These changes partly explain why cadmium exposure and a high cadmium body burden may raise the risk of IA and of disease symptoms exacerbation.

## 1. Introduction

There has been a steady increase in the prevalence of autoimmune diseases worldwide, particularly in the developed, industrialized western and northern countries. Among autoimmune diseases, the prevalence of rheumatic diseases is rising the most, followed by endocrine and gastrointestinal diseases [1]. Rheumatic diseases are an important issue in daily clinical practice; they cause pain, suffering, and disability, and often result in premature retirement. According to the Declaration of the European Parliament on rheumatic diseases [2], musculoskeletal manifestations occur in approximately 30–40% of the European population. The three most common types of inflammatory arthritis (IA) are rheumatoid arthritis (RA, affects nearly 1% of the population worldwide), psoriatic arthritis (PsA), and ankylosing spondylitis (AS). These are chronic, progressive autoimmune diseases whose pathogenesis is still not fully understood [3,4,5]. A common inflammatory pathway for these diseases is the activation of T helper cells 17 (Th17), resulting in an increase in pro-inflammatory cytokines, tumour necrosis factor (TNF-α), and the joint inflammation process [6,7,8]. Genetic factors play an essential role in autoimmunity, but they do not explain all the differences in IA prevalence among populations [9]. As in many other disorders, unfavourable genetic factors are brought forth in specific environmental conditions. Inflammation and various environmental contaminants (immune adjuvants, immunotoxic substances) can induce or exacerbate autoimmunity. Numerous factors have been investigated in association with IA risk, including smoking, diet, physical activity, vitamin D deficiency, alcohol use, medications, and infections [10,11,12].

It is considered that tobacco smoking is significantly associated with RA and PsA risk in the general population; in turn, its impact on AS is linked mainly to disease progression [13,14,15]. The influence of smoking on the autoimmune process has been explained by several hypotheses, involving such factors as increased oxidative stress, general inflammation, and triggering autoantibody production [13,16]. As there is a well-documented link between cigarette smoking and arthritis, a growing interest also has been observed in the role of different environmental pollutants (especially cadmium compounds) in rheumatic diseases [17,18,19]. Cd is a major public health concern, as it is widespread and toxic and contributes to the development of numerous diseases [20,21]. Both in work settings and in daily life, there are many potential sources of cadmium, such as smoking (including passive smoking), vehicle exhaust, low emissions from coal-fired furnaces, recycling of electronic waste, the manufacture of nickel-cadmium batteries, industry (e.g., textiles, dyeing, metallurgy, plastics, mining), as well as contaminated food or cosmetics [22,23,24,25]. The link between cadmium and IA is not clear, having been studied mainly in patients with RA. A Korean cross-sectional study of 53,829 residents found a significant positive association between serum cadmium levels and the incidence of RA [26]. Similar relationships have been observed by other authors [27,28,29,30]. There are indications that exposure to metals (including cadmium) may cause autoimmunity and epigenetic effects, thereby contributing to the incidence and prevalence of autoimmune diseases [31,32]. Some metal compounds modify cytokine production, leading to an imbalance between CD4 Th1 and Th2 cell activation and altering the pro- and anti-inflammatory cytokine profiles. This immune dysregulation can result in impaired cell-mediated immunity and/or aberrant humoral immunity, and, consequently, an autoimmune disease [31]. The influence of cadmium on the development of IA may, therefore, be related to the toxic effect on the immune system and its dysregulation.

This impact on the development of IA can be linked to an anti-pro-oxidative imbalance and the subsequent induction of oxidative stress and inflammatory processes [33]. Cadmium, as a heavy, toxic metal, interferes with the metabolism of biogenic elements. Cadmium inhibits iron absorption by direct suppression of iron transport in duodenal enterocytes [34]. Iron deficiency and anaemia of chronic disease (anaemia of chronic inflammation) are often found in patients with chronic inflammation and autoimmune diseases (including joint diseases) [35,36]. These conditions increase cadmium absorption and accumulation in the body [34]. Low haemoglobin levels are associated with disease activity, joint damage, pain, and functional impairment; they may predict worse outcomes in RA patients [37]. Additionally, decreased zinc levels in the blood are reported in patients with RA. Zinc is a cofactor for more than 3000 proteins and, as a signalling ion, affects many pathways relevant to joint disease. Cadmium absorption increases under zinc deficiency conditions, as in the case of iron. Cadmium dust inhalation can cause macrophage activation, inflammation, and IA development [38]. Rentschler et al. [39] conclude that a genetic variation in zinc transporters influences blood cadmium concentrations (Cd-B): individuals with a certain genotype of these transporters have higher Cd-B. It seems likely that people with this genetic condition are more susceptible to developing the diseases to which cadmium exposure may contribute.

We examined aspects of the potential link between cadmium body burden and the risk of developing IA. Our study aimed to evaluate the relationship between Cd-B and the likelihood of developing and/or exacerbating IA symptoms among patients from the University Clinical Hospital in Wroclaw, Poland. In addition, we used collected data and measured parameters to identify predictors of inflammatory arthritis development in the study population.

Further, our goal was to explore the association between Cd-B and demographic and lifestyle characteristics, selected indicators of inflammatory processes (erythrocyte sedimentation rate (ESR), high-sensitivity C-reactive protein (hs-CRP), cyclooxygenase-2 (COX-2)) and oxidative processes (8-hydroxy-2’-deoxyguanosine (8-OHdG)), as well as factors regulating the immune system (interleukin 10 (IL-10)). The analysis of these relationships may help identify the mechanisms of cadmium effect on the development of autoimmune joint diseases. 

## 2. Materials and Methods

### 2.1. Study Population

This study was conducted at the turn of years 2020/2021 in the University Clinical Hospital in Wroclaw, Poland. Overall, 51 patients who had been hospitalized in the Department of Rheumatology and Internal Medicine were included. The inclusion criteria involved a diagnosis of IA (RA, PsA, or AS), age ≥ 18 years, and consent to participate. Disease activity was evaluated with the 28-joint Disease Activity Score with ESR (DAS28-ESR) in RA patients (<2.6 points—remission; 2.6–3.2—low activity; 3.2–5.1—moderate activity; >5.1—high activity), the Bath Ankylosing Spondylitis Disease Activity Index (BASDAI, 0–10) in AS patients (<2—remission; 2–3—low activity; 3–4—moderate activity; >4—high activity), and Likert score in PsA patients (1—remission; 2—low activity; 3—moderate activity; 4–5—high activity).

The age- and sex-matched control subjects (*n* = 46), not suffering from IA, were patients of the Department of Internal and Occupational Diseases, Hypertension and Clinical Oncology of the same hospital. The study group consisted of patients in a 1-day rheumatology ward who were resistant to routine treatment (nonsteroidal anti-inflammatory drugs, glucocorticoids) and who had been previously or were currently receiving biological therapy or had just been qualified for such treatment. The control group involved patients hospitalized in the internal medicine department (diagnosed in the sleep laboratory) with the aim of verifying the suspicion of obstructive sleep apnoea. They stayed for about a day in the ward for the polysomnographic study. The exclusion criteria for both groups involved neoplastic, mental, and other severe systemic diseases. The subjects completed a short questionnaire concerning demographic characteristics (sex, age, education, employment, chronic diseases) and potential present or past (within the previous 10 years) sources of cadmium exposure (workplace, residence, smoking, leisure activities, diet, use of colour cosmetics). The questionnaire is available in the Appendix A.

The participants provided their written informed consent for inclusion before the study commencement. The study followed the Declaration of Helsinki. The study was approved by the Ethics Committee at Wroclaw Medical University (approval No.: 139/2020) and was registered in the Simple.Bazus system of Wroclaw Medical University under the number SUB.A100.20.019.

### 2.2. Sample Collection

Fasting blood and morning urine were collected from patients in the study and control groups. Blood (ca. 4 mL) was collected from the ulnar vein into two polyethylene terephthalate (PET) plastic tubes (Becton, Dickinson and Company; Franklin Lakes, NJ, USA), with K_2_EDTA to obtain whole blood and without anticoagulant to obtain serum. Serum was prepared by centrifugation (3000× *g*, 10 min) from the blood samples and used for biochemistry analysis. The obtained samples that were not used immediately for the assay were frozen and stored at −70 °C until use.

### 2.3. Determination of Blood Cadmium Content

Cd-B was determined in a certified atomic absorption spectroscopy laboratory of the University Clinical Hospital in Wroclaw, Poland, with the atomic absorption spectroscopy atomization technique in an electrographite cuvette at λ = 228.8 nm, with the use of a SOLAAR M6 (Thermo Elemental Ltd., London, UK) atomic absorption spectrophotometer equipped with a Zeeman background correction system.

The Stoeppler and Brandt [40] method was used for the analysis. Samples were deproteinized with 65% nitric acid for trace metal analysis (Suprapur; Merck KGaA, Darmstadt, Germany). The concentration of cadmium in the sample was established by comparing the absorbance of the sample with a standard curve obtained by using ClinCal Whole Blood Calibrators (Cat. No. 9943) and control samples (ClinChek Whole Blood Controls, level I, II, III; Cat. No. 8840-8843; Recipe Chemicals + Instruments GmbH, Munich, Germany). External laboratory control was performed with the support of the Intercomparison Programme for Toxicological Analyses in Biological Materials (Erlangen, Germany) within the framework of the German External Quality Assessment Scheme (G-EQUAS). The detection limit of the described method was 0.082 µg/L.

### 2.4. Haematological and Biochemical Analysis 

Routine examinations, i.e., blood count with smear, the activity of liver enzymes, creatinine, hs-CRP, and ESR, were performed with automatic analysers in the Central Laboratory of the University Clinical Hospital in Wroclaw. Laboratory analyses of COX-2, IL-10, and 8-OHdG levels were conducted by using a PowerWave XS microplate reader (BioTek Instruments, Winooski, VT, USA). The levels of COX-2 and IL-10 in serum were quantified with the enzyme-linked immunosorbent assay (ELISA) method by using the E0699h and E0056h kits, respectively (EIAab Science Inc., Wuhan, China). For the detection of 8-OHdG in urine (a marker of oxidative damage to DNA), a ready-to-use ELISA-based assay (Cat. No. IM-KOGHS040914E; JaICA, Fukuroi, Japan) was applied. All the above measurements were carried out in accordance with the manufacturer’s recommendations.

### 2.5. Statistical Analysis 

Quantitative variables were expressed as means ± standard deviation. However, in some cases, minimum-maximum ranges, medians, and quartiles were used. Qualitative variables were given as counts and percentages. The distribution and variance of data were verified by the Lilliefors test and Levene’s test, respectively. Parametric data were analysed by the t-test, while the Mann–Whitney U test was applied for non-parametric data. Qualitative variables were analysed with the chi^2^ test (or Yates’ chi^2^ test if at least one cell had an expected count of less than 5). 

In some instances, for analysis purposes, continuous variables were also categorized (e.g., divided into intervals in accordance with the quartile distribution) or transformed into dummy variables (e.g., 0—<median, 1—≥median; or 0—<upper quartile, 1—≥upper quartile). Depending on the distribution, Cd-B values were divided into quartiles as follows: <0.23 μg/L; 0.23 to <0.35 μg/L; 0.35 to <0.63 μg/L. Spearman’s correlation analyses (Kendall’s τ coefficient was used for categorized variables) were conducted to examine the relationship between the measured parameters. 

A binary logistic regression model (Wald test) served to assess the likelihood of IA. The initial step of the analysis was to identify factors significantly influencing the occurrence of IA in the study population. For this purpose, a univariate logistic analysis of potential demographic and laboratory predictors was performed. Only independent variables whose *p*-value in the univariate analysis was less than 0.05 were used to build the model. In turn, the exclusion criterion was an excessive correlation with IA and other prognostic factors. For example, from among red blood cell parameters, only mean corpuscular haemoglobin (MCH) was introduced into the model, while inflammatory markers (ESR and hs-CRP) correlating with cadmium concentration and used to assess disease severity were not introduced. The distribution of the model residuals was checked graphically. As mentioned in Section 2.1, the activity of the studied diseases (RA, AS, PsA) was measured with different scales and indices; therefore, for the purposes of statistical analysis, a scale of disease severity was conventionally adopted as follows: 1—remission, 2—low activity, 3—moderate activity, 4—high activity. 

Correlations between disease activity and cadmium burden, levels of haematological and biochemical parameters, and demographic characteristics in patients with IA were assessed with Kendall’s τ coefficient. The chi^2^ test verified the association between the onset of disease symptoms (remission vs. exacerbation of disease symptoms) and high Cd-B level (≥upper quartile of Cd-B for the study population). 

All statistical analyses were conducted with the Statistica 13.3 software for Windows (StatSoft, Krakow, Poland). Statistical significance was set at *p* < 0.05.

## 3. Results

### 3.1. Basic Demographic Characteristics and Laboratory Parameters 

Basic characteristics and laboratory parameters of the study population are presented in Table 1 and Table 2, respectively. Overall, 97 participants were enrolled in this study: 51 patients with IA (58.8% males) aged 48.1 ± 13.8 years and 46 control subjects (60.9% males) aged 47.5 ± 13.7 years. All respondents were residents of Lower Silesia; 55.2% lived in a large city (Wroclaw), the rest in nearby smaller towns of the region. There were no significant differences in terms of age, sex, place of residence, or education level between the IA patients and controls. Most individuals (88.2% of IA patients and 71.7% of controls) declared environmental exposure to cadmium related to low emissions at home and/or at work. Current smokers accounted for 13.7% of IA patients and 15.6% of controls; 21.6% of IA patients and 31.1% of controls were former smokers; 29.4% of IA patients and 26.7% of controls were exposed to passive smoking. Only 15.7% of IA patients and 8.7% of controls indicated occupational cadmium exposure. Environmental exposure to cadmium as declared by the respondents was similar in both groups. Of the potential sources of cadmium considered, only frequent consumption of offal was reported significantly more often by IA patients than by controls (49% and 24%, respectively, *p* = 0.0085).

Substantial Cd-B levels are associated with long-term exposure, and, like cadmium in urine, reflect cumulative exposure and body burden. However, when exposure varies (e.g., after smoking cessation), the changes occur more rapidly in blood than in urine. For this reason, Cd-B is also used as a measure of current cadmium exposure (previous weeks and months) [41,42,43]. Mean Cd-B was significantly higher in patients with IA (0.670 ± 0.513 μg/L) than among controls (0.40 ± 0.47 μg/L).

As shown in Table 2, white blood cell count, mainly neutrophil and monocyte count, was increased in IA patients compared with controls. The inflammatory markers (ESR, hs-CRP, COX-2) were higher in IA subjects. In addition, haemoglobin levels and related red cell parameters (red blood cells, haematocrit (HCT), MCH, and mean corpuscular haemoglobin concentration (MCHC)) were statistically significantly reduced in IA patients compared with controls. Over 12% of patients and only 4% of controls had blood count results suggestive of iron deficiency anaemia (data not shown). IL-10 levels were significantly higher in the IA group. Urinary 8-OHdG levels (oxidative DNA damage indicator) were higher in IA patients than in controls (12.02 vs. 7.03 ng/mL), but these differences were not statistically significant. Nevertheless, 8-OHdG concentrations were positively correlated with serum hs-CRP levels (*r* = 0.22, *p* = 0.034, not shown in the table).

### 3.2. Correlations between Blood Cadmium Concentrations and Basic Characteristics

Correlations between potential sources of cadmium, some basic characteristics of the study population, and Cd-B are presented in Table 3. Cd-B correlated positively with age (*r* = 0.25, *p* < 0.05), current (*r* = 0.46, *p* < 0.01), past (*r* = 0.36, *p* < 0.01), and passive (*r* = 0.29, *p* < 0.01) smoking, as well as living in a high-traffic area (*r* = 0.32, *p* < 0.01). A negative correlation was observed with education level (*r* = –0.20, *p* < 0.05).

As shown in Table 4, Cd-B correlated positively with ESR (*r* = 0.27, *p* < 0.01), hs-CRP (*r* = 0.24, *p* < 0.05), and COX-2 (*r* = 0.30, *p* < 0.01). A negative correlation was observed with MCHC (*r* = −0.34, *p* < 0.01).

### 3.3. Logistic Regression Model

The selection of predictors to create the model was based on univariate logistic analysis. Table 5 shows exemplary results of a univariate logistic analysis assessing IA probability in relation to Cd-B (considered as a categorized variable, divided into 4 intervals depending on the quartile distribution). Elevated Cd-B was significantly associated with the risk of IA, with an odds ratio (OR) of 1.9 (95% confidence interval [CI]: 1.3–2.8, *p* = 0.00083), for a 1 interval increase in cadmium concentration.

As mentioned in Section 2.5, the choice of the optimal set of variables for the final model was performed by using univariate logistic analysis, followed by backward stepwise analysis. 

The model involved factors related to cadmium exposure (Cd-B), diet, red blood cell parameters, and markers of oxidative stress and inflammatory processes (the inflammatory marker COX-2 was not considered in the clinical assessment of IA activity). The binary logistic regression analysis resulted in the final model containing the following explanatory variables: Cd-B ≥ 0.65 μg/L, 8-OHdG ≥ 7.66 ng/mL (≥median), COX-2 ≥ 22.9 ng/mL, MCH (pg), and frequent consumption of offal (≥once a week). Details of the regression parameter estimates are shown in Table 6.

The obtained results point to an increased probability of high values of Cd-B (≥0.65 μg/L), 8-OHdG (≥7.66 ng/mL), and COX-2 (≥22.9 ng/mL), as well as frequent consumption of offal and a decreased MCH level in individuals suffering from IA. This suggests that increased Cd-B (≥0.65 μg/L) may raise the risk of IA occurrence. These results also imply that high concentrations of 8-OHdG (≥7.66 ng/mL) and COX-2 (≥22.9 ng/mL), frequent consumption of offal, and a decreased MCH level can be associated with an increased risk of autoimmune joint diseases.

The final model considering the respective parameters for each predictor variable could be presented as follows:P(x) = e ^9.9 + 1.49 * Cd-B (≥0.65 μg/L) + 2.36 * 8-OHdG (≥7.66 ng/mL) + 1.76 * COX−2 (≥22.9 ng/mL) − 0.39 * MCH + 1.49 * frequent consumption of offal^/1 − e ^9.9 + 1.49 * Cd-B (≥0.65 μg/L) + 2.36 * 8-OHdG (≥7.66 ng/mL) + 1.76 * COX−2 (≥22.9 ng/mL) − 0.39 * MCH + 1.49 * frequent consumption of offal^(1)
where P(x)—probability of IA in the study population.

The respective OR values for high levels of Cd-B (≥0.65 μg/L), 8-OHdG (≥7.66 ng/mL), COX-2 (≥22.9 ng/mL), and frequent offal consumption equalled: 4.4 (95% CI: 1.1–18.4, *p* = 0.04), 10.6 (95% CI: 2.7–40.7, *p* = 0.001), 5.8 (95% CI: 1.4–24.2, *p* = 0.016), and 3.6 (95% CI: 1.2–11.1, *p* = 0.025). In turn, the OR value for MCH increase by 1 pg in IA patients was 0.7 (95% CI: 0.5–0.9, *p* = 0.022), which implies an association between MCH decrease and the development of autoimmune joint diseases. The logistic regression model was statistically significant and correctly classified 80.39% of IA patients and 80.43% of controls; the overall classification success rate was 80.41%. The chi^2^ goodness of fit test results (chi2 = 49.328, df = 5, *p* = 0.0000000) confirmed the good fit of the model.

### 3.4. Relationship between Disease Severity and Demographic Characteristics, Laboratory Parameters, and Cadmium Body Burden

The correlations between disease activity (disease severity scale) and demographic and laboratory characteristics of IA patients are depicted in Table 7.

We observed a negative correlation between the severity of disease symptoms and red blood cell indices such as haemoglobin, HCT, mean corpuscular volume, MCH, and MCHC (*r* = −0.19, *p* < 0.05; *r* = –0.20, *p* < 0.05; *r* = –0.20, *p* < 0.05; *r* = −0.22, *p* < 0.05, and *r* = −0.3, *p* < 0.01, respectively), and a positive relationship for platelet count (*r* = 0.23, *p* < 0.05). As expected, serum markers of inflammation (ESR and hs-CRP) were positively related to the disease severity, both when considered as continuous variables (*r* = 0.39, *p* < 0.01; *r* = 0.3, *p* < 0.01, respectively) and categorized (*r* = 0.27, *p* < 0.01; *r* = 0.38, *p* < 0.01, respectively).

An association between more severe disease symptoms and higher education (*r* = 0.27, *p* < 0.01) and female sex (*r* = 0.19, *p* < 0.05) was noted; in turn, the correlation of symptom severity and disease duration was negative (*r* = 0.21, *p* < 0.05).

Moreover, we established that the highest Cd-B (≥upper quartile) and 8-OHdG urine levels (≥upper quartile) were related with more severe disease symptoms (*r* = 0.27, *p* < 0.01; *r* = 0.38, *p* < 0.01, respectively). A relationship between a decline in serum IL-10 levels and increased severity of disease symptoms was also observed (*r* = 0.51, *p* < 0.01).

Figure 1 (interaction graph) illustrates the relationship between the exacerbation of disease symptoms in IA patients and high Cd-B (≥0.65 μg/L).

More than 84% of IA patients whose Cd-B was equal to or higher than the upper quartile value for the study population (0.65 μg/L) faced disease symptom exacerbation, whereas remission occurred in 15.8% only. These dependencies were statistically significant (Yates’ chi^2^ = 3.91, *p* < 0.05; *r* = 0.375, *p* < 0.05).

Patients with elevated Cd-B (≥0.65 μg/L) were 6-fold more likely to experience onset of disease symptoms (OR = 6.09, 95% CI: 1.5–25.4, *p* < 0.05) than those with lower Cd-B (<0.65 μg/L).

Figure 2 shows the values of serum IL-10 in IA patients and controls depending on Cd-B. The serum IL-10 level in IA patients was higher than in the control group—both with high (≥0.65 μg/L, ≥upper quartile) and low (<0.65 μg/L) Cd-B values. However, IA patients with high Cd-B had significantly decreased IL-10 levels compared with those with low Cd-B (*p* = 0.05). In the control group, IL-10 was higher in individuals with elevated Cd-B, but the difference was not statistically significant. 

## 4. Discussion

The mean Cd-B of the study population from the University Clinical Hospital in Wroclaw (Lower Silesia, Poland) was 0.54 ± 0.48 μg/L (range: 0.04–2.16 μg/L, geometric mean: 0.37 μg/L) and was similar to that observed in adults from developed countries in the world (geometric mean: 0.31 μg/L in Sweden, 0.34 μg/L in the USA, 0.4 μg/L in Canada) [44,45,46].

We noted, however, that Cd-B values in IA patients were higher than in control subjects (mean ± standard deviation: 0.67 ± 0.51 vs. 0.40 ± 0.47 µg/L, *p* < 0.01).

Similar results were obtained by Joo et al. [26] in a population-based cross-sectional study in Korea involving 53,829 participants. Of the heavy metals analysed by these authors (cadmium, lead, mercury, manganese, and zinc in serum, arsenic in urine), only serum cadmium concentrations were significantly higher in patients with RA than in controls. However, these differences were observed only in women. The prevalence of RA in women increased with rising quartiles of serum cadmium levels and was 19-fold higher in individuals in the highest quartile than among those in the lowest quartile. Hutchinson [28] put forward a hypothesis that seropositive RA (the most common RA form) was an environmentally dependent disorder. It is indicated by the very low concordance of the disease in monozygotic twins and the fact that it appeared and was first described as late as in the early 19th century. The disease did not occur before the industrial revolution when the concentration of cadmium in the environment was still relatively low. Hence, it seems likely that cadmium could play a part in both RA severity and susceptibility. This may also apply to other autoimmune joint diseases.

As we expected, cigarette smoking had the most pronounced effect on blood cadmium levels. We observed an association between current, past, as well as passive smoking and elevated Cd-B. The standpoint that smoking has the strongest effect on cadmium body burden in the general population is widely accepted. Cadmium is the main contributor to the health risks associated with metals in tobacco smoke [47,48]. Mechanisms of cadmium toxicity such as disruption of biogenic element metabolism, DNA damage repair inhibition, induction of oxidative stress and inflammatory processes, among others, have been described [49,50,51]. According to some authors, cadmium causes DNA methylation and may affect smoking toxicity also through epigenetic mechanisms [23,52]. All of these processes may be involved in cadmium’s contribution to the development of autoimmune joint diseases.

Among the potential sources of cadmium considered, only frequent consumption of offal was reported significantly more often by IA patients than by controls.

Food is the main source of cadmium exposure in non-smokers and those not occupationally exposed to the metal or its compounds. Dietary sources of cadmium vary by geographic conditions, population groups, and individual consumer preferences. In some countries, seafood and organ meats (e.g., kidney, liver) add to the highest dietary intake of cadmium [22,53]. In Poland, the consumption of fish and seafood is low [54], but some offal is eaten readily. Offal is considered a lower-quality product and is now less frequently consumed than several decades ago when it was a substitute for meat owing to market shortages. However, offal has its admirers and is treated by them as a delicacy. Therefore, there can be considerable differences in offal intake. In Poland, among offal products, the liver is frequently consumed (poultry, pork, and beef) [55,56]. The liver is a source of complete protein, vitamins, and minerals [54]; however, because of its function in the body, it also accumulates heavy metals and toxins. High levels of cadmium are detected in edible offal, especially in kidneys and liver [53,57,58,59]. The higher consumption of offal (liver) among the IA patients as compared with the controls may be explained by the fact that it is generally perceived as healthy. It is widely believed that its consumption can improve blood parameters and correct iron deficiency in individuals with anaemia.

Another cause of elevated Cd-B values in IA patients might be intensified absorption of this metal. Decreased levels of biogenic metals such as iron and zinc increase the absorption of cadmium [34,39,60]. This may be due to genetic conditions. For example, individuals with a certain genotype of zinc transporters (SLC39A8 and SLC39A14) absorb larger quantities of cadmium [39]. It is likely that such people are more susceptible to developing diseases to which cadmium exposure may contribute.

Several authors have found zinc and iron deficiencies in patients with autoimmune joint diseases [38,61]. This could also be the case in our subjects. Although we did not determine these trace elements in the study, we observed lowered haemoglobin levels and changes in red blood cell parameters in IA patients. These alterations may indicate a reduced iron status. We revealed that decreased MCHC correlated with increased Cd-B. Smyrnova [37] implied that a low haemoglobin level was related to disease activity, disability, articular damage, pain, and disease duration in patients with RA. The study reported a correlation between the severity of disease symptoms and the decline of such red blood cell parameters as haemoglobin, HCT, mean corpuscular volume, MCH, and MCHC. The viewpoint that cadmium exposure can exacerbate autoimmunity in genetically predisposed individuals and play a role in autoimmune disease development is increasingly accepted by researchers [62]. Future studies to identify factors contributing to IA development should consider relationships between trace element deficiencies (including those resulting from genetic conditions) and increased cadmium or other heavy metal uptake.

Levels of inflammatory markers, particularly C-reactive protein, positively correlate with disease activity and joint damage in IA patients [63]. In turn, increased production of COX-2 and prostaglandin E2 by chondrocytes enhances the degradation of both aggrecan (cartilage-specific protein, critical for cartilage structure and the function of joints) and collagen type II in patients with osteoarthritis [64]. It can thus be expected that an increase in the inflammatory processes induced by cadmium exposure will favour the development of inflammatory joint diseases.

In our study, we noted that white blood cell count, mainly neutrophil and monocyte count, was increased in IA patients compared with controls. This was an expected finding. Reports from recent years indicate the critical roles of neutrophils and monocytes/macrophages during an aberrant immune response. Both of these cell types are capable of driving various autoimmune and inflammatory diseases [65]. The inflammatory markers (ESR, hs-CRP, COX-2) were also higher in IA patients. We demonstrated significant positive correlations between Cd-B and serum levels of inflammatory markers. At the same time, as expected, inflammatory markers exhibited a positive correlation with disease severity in IA patients. These relationships indicate that an increased body burden of cadmium may influence the progression of inflammatory processes in autoimmune joint diseases.

There are various studies regarding the effects of cadmium on inflammation induction and the resultant development of numerous diseases [62,66,67]. Xiao et al. [68] observed significant positive dose-response relationships between urinary cadmium and plasma C-reactive protein, and between plasma C-reactive protein and type 2 diabetes risk. Exposure to cadmium could increase COX-2 by activating the mitogen-activated protein kinase pathway. In turn, cadmium-induced COX-2 elevation increases the production of inflammatory prostaglandin E2, leading to damage to endothelial vascular cells [67]. An analysis of data from 38,223 participants in the National Health and Nutrition Examination Survey (NHANES) conducted by Ma et al. [69] also indicates that cadmium may increase the risk of cardiovascular disease by elevating blood lipids and promoting inflammation. The authors demonstrated that raised cadmium concentrations increased the levels of inflammatory factors, such as white blood cells and C-reactive protein. Cadmium induces inflammation in the kidneys and liver, leading to dysfunction and damage. The inflammatory processes provoked by cadmium are closely related to oxidative stress in the tissues and disturbance of the redox balance. Excessive generation of reactive oxygen species activates the transcription of NF-κB, resulting in the overproduction of pro-inflammatory mediators [66,70,71]. Larson-Casey et al. [72] reported that cadmium induces a rapid and persistent polarization of lung macrophages to a pro-inflammatory phenotype. Generated by cadmium, reactive oxygen species in the mitochondria regulate redox-regulated transcription factors to maintain the persistence of the pro-inflammatory phenotype; these changes exacerbate lung injury.

The univariate logistic analysis indicated that increased Cd-B was significantly associated with the risk of IA, with an OR of 1.9 (95% CI: 1.3–2.8, *p* = 0.00083) for a one quartile interval increase in cadmium concentration. We also observed that in patients diagnosed with IA, high Cd-B (≥0.65 μg/L, upper quartile) was significantly associated with both an onset (OR = 6.09, 95% CI: 1.5–25.4, *p* < 0.05) and exacerbation (*r* = 0.23, *p* < 0.05) of disease symptoms.

The findings suggest that growing Cd-B increases inflammatory processes, promoting the onset of IA symptoms and escalation of the disease activity. Our results are in line with the viewpoints presented in the current literature. Namely, elevated exposure to metal compounds can modify cytokine production and alter the profile of pro- and anti-inflammatory factors. It may impair cell-mediated and/or humoral immunity and, consequently, lead to the manifestation and/or exacerbation of autoimmune disease symptoms [31]. There are reports indicating that cadmium exposure increases serum levels of pro-inflammatory cytokines [66,67]. Additionally, cadmium-induced inflammatory processes are exacerbated by the down-regulation of anti-inflammatory IL-10 [73].

IL-10 is an important immunomodulatory cytokine. Its essential functions include inhibiting the production of pro-inflammatory cytokines and mitigating the effects of inflammation. Its therapeutic potential in autoimmune diseases (including IA) has been demonstrated [74,75]. IL-10 modulates the biological activity of immune cells. It potently inhibits the activation of these cells, reducing the production of pro-inflammatory mediators, such as cytokines and chemokines, and decreasing T-lymphocyte stimulation. The inhibitory effects of IL-10 on the pro-inflammatory functions of granulocytes, Th cells, NK cells, and endothelial cells give it a main role as a ‘damper’ of inflammation. IL-10 protects tissues from the impact of strong inflammatory reactions, which is essential in alleviating symptoms of autoimmune inflammatory diseases [75,76]. The pathogenesis of arthritis strongly depends upon the secretion of pro-inflammatory cytokines and the subsequent recruitment of inflammatory cells. IL-10, producing regulatory B (Breg) cells, restrains inflammation by promoting differentiation of immunoregulatory and pro-inflammatory T cells and is responsible for controlling autoimmunity. In chimeric mice that had IL-10 knocked-out, specifically in the B cell population, a marked increase in inflammatory Th1 and Th17 cells and enhanced susceptibility to collagen-induced arthritis was observed compared with wild-type B cell chimeric mice. Joint damage was more severe in mice lacking IL-10, specifically in their B cells, than in those with wild-type B cells [77]. Similarly, Greenhill et al. [78] demonstrated that IL-10 deficiency increased the degree of bone erosion observed in IA and aggravated the disease course. The level of circulating IL-10 can be influenced by many factors, including exposure to metals [79,80,81], exercise [82], receiving substances of anti-inflammatory activity [79], and type of diet [83]. These modifications may exacerbate or relieve inflammatory joint disease symptoms.

We noted that the serum IL-10 level was higher in IA patients than among controls. This corroborates literature reporting that a rise in IL-10 levels in patients with inflammatory diseases (such as RA) is involved in diminishing the disease activity. Unfortunately, the therapeutic application of IL-10 is limited. IL-10 can promote humoral immune responses, enhancing class II expression on B cells and inducing immunoglobulin production. After serial administration of high doses of IL-10, clinical complications were observed such as neutrophilia, monocytosis, and lymphopenia. Hence, developing safe IL-10 therapies appears to be important [84].

In our study, the severity of IA symptoms showed a strong inverse relationship with serum IL-10 levels. This supports the thesis that a rise in IL-10 levels alleviates symptoms of inflammatory disease. However, IA patients with high Cd-B levels (≥0.65 μg/L) had lower IL-10 concentrations than those with lower Cd-B levels (<0.65 μg/L). This implies that cadmium exposure may also exacerbate symptoms of autoimmune joint diseases by decreasing the anti-inflammatory IL-10.

Some studies confirm that cadmium toxicity is due to the down-regulation of cytokine expression. Odewumi et al. [81] revealed that in normal lung cells, cadmium affected the expression levels of cytokines, which was related to its cytotoxic effects in the lung. The reduction in IL-10 levels was influenced by higher cadmium concentrations and longer exposure times. Akinyemi and Adeniyi [79] reported that cadmium administration caused nephrotoxicity in rats by a significant IL-10 level reduction and elevation in pro-inflammatory cytokines (IL-6 and TNF-α). However, co-treatment with essential oil from ginger and turmeric rhizome exhibited anti-inflammatory and neuroprotective activity by preventing a decrease in IL-10 and reducing IL-6 and TNF-α. Therefore, it appears that factors that influence inflammatory processes may modulate cadmium toxicity. We consider that reduced levels of IL-10 are partially responsible for more severe inflammation and more active disease in IA patients with high Cd-B compared with those with lower Cd-B levels. A new observation in our study appears to be the demonstration of a difference in the effect of high blood cadmium concentrations on anti-inflammatory IL-10 levels in the IA patients and controls. High Cd-B (≥upper quartile) significantly reduced IL-10 levels in the IA patients and had no effect on IL-10 or even increased its levels (not statistically significantly) in the control group. This phenomenon may be related to a higher sensitivity to cadmium in individuals who are predisposed to the development of autoimmune joint diseases.

We applied binary logistic regression analysis to identify factors related to IA probability in the study population. A high Cd-B concentration (≥0.65 μg/L) increased the probability of IA occurrence more than 4-fold. A higher risk of the disease was also associated with high levels of 8-OHdG (≥7.66 ng/mL) and COX-2 (≥22.9 ng/mL), as well as with more frequent consumption of offal. In turn, an increase in MCH values was accompanied by a lower IA risk. However, among the disease predictors included in the final binary logistic regression model, high urinary 8-OHdG concentration (≥7.66 ng/mL, ≥median) had the greatest effect, increasing the probability of IA occurrence more than 10-fold (OR = 10.6, 95% CI: 2.7–40.7, *p* = 0.001).

Oxidative stress is of major significance in both cadmium toxicity [51,72] and the development of autoimmune diseases [85,86]. The 8-OHdG level modification is the most common change caused by the reaction between hydroxyl radicals and guanosine in DNA and constitutes a critical biomarker of oxidative stress, mutagenesis, carcinogenesis, and degenerative disease development [87]. A high predictive value of 8-OHdG has been demonstrated in RA [85].

In our study, urinary 8-OHdG levels were higher in IA patients than in controls (12.02 vs. 7.03 ng/mL), but the differences were not statistically significant. Neither did we find a statistically significant correlation between Cd-B and urinary 8-OHdG levels. Nevertheless, urinary 8-OHdG levels were positively correlated with serum hs-CRP concentration (*r* = 0.22, *p* = 0.034), a sensitive marker of inflammation, used in diagnosing IA.

The lack of correlation between Cd-B and urinary 8-OHdG levels may be explained by the fact that oxidative DNA damage is associated with failure of repair processes and may represent a more distant change, whereas Cd-B is rather indicative of current exposure. Similarly, a study by Xu et al. [88] on health disorders in people living near electroplating facilities revealed no significant correlation between 8-OHdG and Cd-B. However, a statistically significant relationship was observed between 8-OHdG levels and urinary cadmium concentrations, which mainly reflect long-term exposure.

There are several limitations to our study. The relatively small sample size prevented a stratification analysis and demonstration of associations between Cd-B and IA development depending on joint disease type, sex, age, etc. We also did not evaluate the influence of the administered medications on the determined parameters in either the IA or the control group. Moreover, the control group consisted of hospital patients who, despite the initial selection, were burdened with various (although not severe) ailments and diseases; this may have reduced the reliability of the differences reported between the groups.

## 5. Conclusions

Cadmium induces and/or exacerbates processes that contribute to the development of autoimmune joint diseases. These processes include oxidative stress, disturbances in the metabolism of biogenic elements such as zinc and iron, or inflammation. In our study, Cd-B correlated positively with serum markers of inflammatory processes and negatively with haemoglobin levels. These parameters are indicators in diagnosing inflammatory joint diseases or, as in the case of reduced haemoglobin levels, often accompany these diseases. High Cd-B (upper quartile) was significantly associated with an increased likelihood of autoimmune joint disease in the investigated population. In IA patients, high Cd-B was associated with reduced anti-inflammatory cytokine IL-10 levels and exacerbated disease symptoms.

The results indicate that high Cd-B increases the likelihood of IA occurrence, as well as of symptom relapse and exacerbation in diagnosed patients. On the basis of the conducted questionnaire interview, no significant differences in cadmium exposure could be found between IA patients and the control group. Of the potential sources of cadmium considered, only offal consumption frequency was higher in IA patients than in the controls. It seems likely that the raised Cd-B in IA patients may be related to increased absorption of this metal. Further studies should clarify whether increased cadmium uptake, e.g., genetically determined, predisposes to autoimmune joint diseases.

## Figures and Tables

**Figure 1 ijerph-19-03099-f001:**
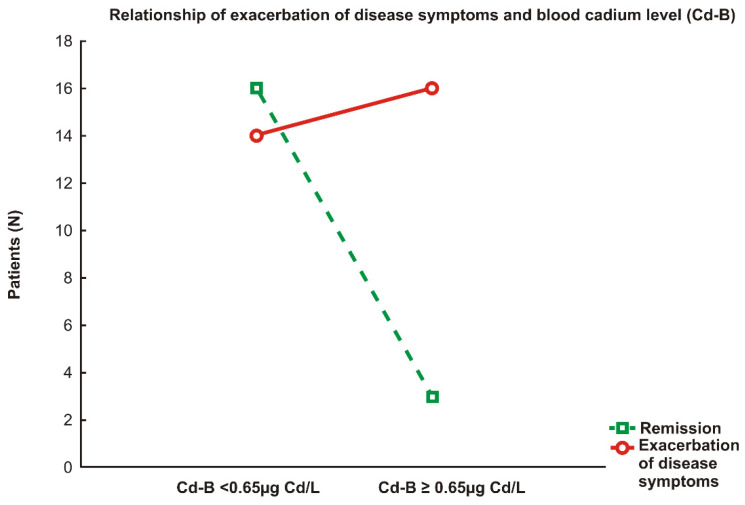
Relationship between the exacerbation of inflammatory arthritis (IA) symptoms and high blood cadmium concentrations (Cd-B ≥ 0.65 μg/L).

**Figure 2 ijerph-19-03099-f002:**
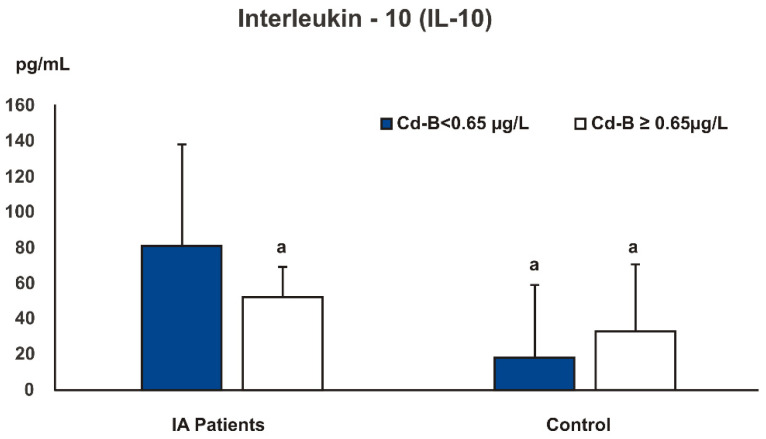
Serum interleukin 10 in inflammatory arthritis patients and controls depending on blood cadmium concentrations (Cd-B). ^a^ significant difference (*p* < 0.05) versus patient group with lower blood cadmium levels, Cd-B < 0.65 μg/L.

**Table 1 ijerph-19-03099-t001:** Exposure to cadmium and demographic and other basic characteristics of the study population.

Characteristics	IA Patients (RA, PsA, AS) *n* = 51	Controls *n* = 46	Overall *n* = 97	*p* ^1^
Age (years), mean ± SD	48.1 ± 13.8	47.5 ± 13.7	47.8 ± 13.6	n.s.
Gender male, *n* (%)	30 (58.8%)	28 (60.9%)	58 (59.8%)	n.s.
Place of residence				
Big city, *n* (%)	24 (47.1%)	29 (64.4%)	53 (55.2%)	n.s.
Education level				
Higher, *n* (%)	26 (51%)	23 (51%)	49 (51%)	n.s.
Potential sources of cadmium				
Exposure to low-pollutant emissions (burning coal and rubbish in domestic furnaces, living/working in a heavy-traffic zone), *n* (%)	45 (88.2%)	33 (71.7%)	78 (80.4%)	n.s.
Diet				
Frequent consumption of offal (≥once a week),				
*n* (%)	25 (49%)	11 (24%)	36 (37.5%)	0.0085
Frequent consumption of legumes				
(≥once a week) *n* (%)	25 (49%)	19 (41.3%)	44 (45.4%)	n.s.
Using colour cosmetics (lipstick, powder, eye shadows), *n* (% of females)	16 (76.2%)	9 (50%)	25 (64.1%)	n.s.
Smoking status				n.s.
Never smoked, *n* (%)	33 (64.7%)	24(53.3%)	57 (59.4%)	
Former smoker, *n* (%)	11 (21.6%)	14 (31.1%)	25 (26.0%)	
Current smoker, *n* (%)	7 (13.7%)	7 (15.6%)	14 (14.6%)	
Passive smoker, *n* (%)	15 (29.4%)	12 (26.7%)	27 (28.1%)	n.s.
Occupational exposure to Cd, *n* (%)	8 (15.7%)	4 (8.7%)	12 (12.4%)	n.s.
Disease duration (years), mean ± SD	4.5 ± 6.6			
Disease, *n* (%)				
RA	21 (41.2%)			
AS	17 (33.3%)			
PsA	13 (25.5%)			
Disease activity				
Remission, *n* (%)	20 (39.2%)			
Exacerbation of disease symptoms, *n* (%)	31 (68.0%)			
Low	13 (25.5%)			
Moderate	5 (5.0%)			
High	13 (25.5%)			
Blood cadmium concentration				
Mean ± SD (μg/L)	0.67 ± 0.51	0.40 ± 0.47	0.543 ± 0.48	0.00003
Geometric mean (μg/L)	0.532	0.258	0.373	
Range (min–max) (μg/L)	(0.13–2.16)	(0.00–2.15)	(0.00–2.16)	

IA—inflammatory arthritis, RA—rheumatoid arthritis, PsA—psoriatic arthritis, AS—ankylosing spondylitis, n.s.—not statistically significant. ^1^ The *t*-test or Mann–Whitney *U* test (for continuous variables) and chi^2^ test (for categorical variables) were used to estimate significant differences (*p* < 0.05) between IA patients and control.

**Table 2 ijerph-19-03099-t002:** Laboratory parameters of IA patients and controls (mean ± SD).

Parameters	IA Patients (RA, PsA, AS) *n* = 51	Controls *n* = 46	Overall *n* = 97	*p* ^1^
WBC (×10^3^/μL)	7.74 ± 2.28	6.19 ± 1.52	7.01 ± 2.1	0.0003
Lymphocytes (×10^3^/μL)	1.93 ± 0.69	2.01 ± 0.63	1.97 ± 0.66	n.s.
Neutrophils (×10^3^/μL)	4.90 ± 2.22	3.39 ± 1.11	4.19 ± 1.93	0.0001
Monocytes (×10^3^/μL)	0.67 ± 0.22	0.59 ± 0.49	0.63 ± 0.37	0.001
Haemoglobin (g/dL)	13.75 ± 1.68	15.02 ± 1.45	14.34 ± 1.69	0.0001
RBC (×10^3^/μL)	4.67 ± 0.49	4.92 ± 0.48	4.79 ± 0.50	0.01
HCT (%)	40.37 ± 7.45	43.95 ± 3.92	41.73 ± 6.55	0.0006
MCV (fL)	89.29 ± 4.95	90.32 ± 4.69	89.98 ± 4.83	n.s.
MCH (pg)	29.56 ± 2.26	30.50 ± 1.54	30.00 ± 2.00	0.02
MCHC (g/dL)	32.70 ± 1.25	33.74 ± 0.93	33.10 ± 1.24	0.0004
PLT (×10^3^/μL)	287.8 ± 98.3	247.2 ± 63.7	273.2 ± 95.4	n.s.
AST (U/L)	26.75 ± 11.24	25.59 ± 7.61	26.31 ± 9.97	n.s.
ALT (U/L)	29.45 ± 22.73	27.81 ± 14.63	28.83 ± 19.94	n.s
CRE (mg/dL)	0.93 ± 0.19	0.98 ± 0.15	0.95 ± 0.17	0.03
ESR (mm/h)	16.8 ± 19.6	8.07 ± 7.7	12.7 ± 15.8	0.005
hs-CRP (mg/L)	11.03 ± 26.8	2.30 ± 3.6	7.00 ± 20.2	0.005
COX-2 (ng/mL)	18.46 ± 11.8	9.90 ± 11.91	14.36 ± 12.57	0.00005
8-OHdG (ng/mL)	12.02 ± 12.6	7.03 ± 3.2	9.8 ± 9.8	n.s.
IL-10 (pg/mL)	69.87 ± 52.90	19.49 ± 20.4	45.72 ± 47.76	0.000001

IA—inflammatory arthritis, RA—rheumatoid arthritis, PsA—psoriatic arthritis, AS—ankylosing spondylitis, WBC—white blood cells, RBC—red blood cells, HCT—haematocrit, MCV—mean corpuscular volume, MCH—mean corpuscular haemoglobin, MCHC—mean corpuscular haemoglobin concentration, PLT—platelets, AST—aspartate aminotransferase, ALT—alanine aminotransferase, CRE—creatinine, ESR—erythrocyte sedimentation rate, hs-CRP—high-sensitivity C-reactive protein, COX-2—cyclooxygenase-2, 8-OHdG—8-hydroxy-2’-deoxyguanosine, IL-10—interleukin 10, n.s.—not statistically significant. ^1^ The *t*-test or Mann–Whitney U test (for nonparametric data) was used to estimate significant differences (*p* < 0.05) between IA patients and control group.

**Table 3 ijerph-19-03099-t003:** Correlations between potential sources of cadmium, basic characteristics of the study population, and Cd-B.

	Age (Years)	Level of Education	Current Smoking	Smoking in the Past	Passive Smoking	Living in a Heavy-Traffic Zone
Cd-B	*r* = 0.25 *	*r* = –0.20 *	*r* = 0.46 **	*r* = 0.36 **	*r* = 0.29 **	*r* = 0.32 **

Cd-B—blood cadmium concentration, *r*—Spearman’s rank correlation coefficient. * *p* < 0.05, ** *p* < 0.01.

**Table 4 ijerph-19-03099-t004:** Correlations between Cd-B and laboratory parameters in the study population.

	ESR	hs-CRP	MCHC	COX-2
Cd-B	*r* = 0.27 **	*r* = 0.24 *	*r* = –0.34 **	*r* = 0.30 **

Cd-B—blood cadmium concentration, ESR—erythrocyte sedimentation rate, hs-CRP—high-sensitivity C-reactive protein, MCHC—mean corpuscular haemoglobin concentration, COX-2—cyclooxygenase-2, *r*—Spearman’s rank correlation coefficient.* *p* < 0.05, ** *p* < 0.01.

**Table 5 ijerph-19-03099-t005:** Results of univariate logistic analysis assessing the effect of Cd-B on the probability of inflammatory arthritis in the study population.

	Estimate (B)	*p*	Wald chi^2^	Wald *p*	OR	95% CI
Cd-B (quartile intervals)	0.646	0.001	11.17	0.0008	1.9	1.3–2.8

Cd-B—blood cadmium concentration, ESR—erythrocyte sedimentation rate, hs-CRP—high-sensitivity C-reactive protein, MCHC—mean corpuscular haemoglobin concentration, COX-2—cyclooxygenase-2, *r*—Spearman’s rank correlation coefficient.

**Table 6 ijerph-19-03099-t006:** Binary logistic regression model of selected inflammatory arthritis predictors in the study population (patients vs. controls).

Variables	Estimate (B)	*p*	Wald chi^2^	Wald *p*	OR	95% CI
Constant (B_0_)	9.90	0.046	4.08	0.043	1.99 × 10^4^	1.2–3.4 × 10^8^
Cd-B ≥ 0.65 μg/L (≥upper quartile)	1.49	0.040	4.32	0.038	4.4	1.1–18.4
8-OHdG ≥ 7.66 ng/mL (≥median)	2.36	0.001	12.05	0.001	10.6	2.7–40.7
COX-2 ≥ 22.9 ng/mL (≥upper quartile)	1.76	0.016	6.01	0.014	5.8	1.4–24.2
MCH (pg)	–0.39	0.022	5.47	0.019	0.7	0.5–0.9
Frequent consumption of offal (≥once a week)	1.29	0.025	5.19	0.023	3.6	1.2–11.1

Cd-B—blood cadmium concentration, 8-OHdG—8-hydroxy-2’-deoxyguanosine, COX-2—cyclooxygenase-2, MCH—mean corpuscular haemoglobin, OR—odds ratio, 95% CI-95% confidence interval.

**Table 7 ijerph-19-03099-t007:** Correlations between disease activity (severity of disease scale) and demographic and laboratory parameters in IA patients.

Parameters	Disease Severity Rating Scale
Gender (female)	*r* = 0.19 *, *p* = 0.045
Level of education (higher)	*r* = 0.27 **, *p* = 0.004
ESR	*r* = 0.39 **, *p* = 0.00005
ESR (norm–0, above the norm–1)	*r* = 0.27 **, *p* = 0.006
hs-CRP	*r* = 0.30 **, *p* = 0.002
hs-CRP (norm–0, above the norm–1)	*r* = 0.38 **, *p* = 0.00009
HB	*r* = −0.19 *, *p* = 0.044
HCT	*r* = −0.20 *, *p* = 0.036
PLT	*r* = 0.23 *, *p* = 0.015
MCV	*r* = −0.20 *, *p* = 0.039
MCH	*r* = −0.22 *, *p* = 0.019
MCHC	*r* = −0.30 **, *p* = 0.002
Creatinine	*r* = −0.27 **, *p* = 0.005
Disease duration (years)	*r* = −0.21 *, *p* = 0.044
IL-10	*r* = −0.51 **, *p* = 0.0000
8-OHdG ≥ 10.1 ng/mL (≥upper quartile–1)	*r* = 0.31 *, *p* = 0.015
Cd-B ≥ 0.65 μg/L (≥upper quartile–1)	*r* = 0.23 *, *p* = 0.016

IA—inflammatory arthritis, ESR—erythrocyte sedimentation rate, hs-CRP—high-sensitivity C-reactive protein, HB—haemoglobin, HCT—haematocrit, PLT—platelets, MCV—mean corpuscular volume, MCH—mean corpuscular haemoglobin, MCHC—mean corpuscular haemoglobin concentration, IL-10—interleukin 10, 8-OHdG—8-hydroxy-2′-deoxyguanosine, Cd-B—blood cadmium concentration, *r*—Kendall’s τ correlation coefficient. * *p* < 0.05, ** *p* < 0.01.

## Data Availability

Data are available in patients; records at University Clinical Hospital in Wroclaw, Poland.

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
