# Peer review of "Cadmium Body Burden and Inflammatory Arthritis: A Pilot Study in Patients from Lower Silesia, Poland"

_ijerph, 2022, doi:10.3390/ijerph19053099_

Round 1
Reviewer 1 Report
This is cross-sectional study of a small convenience sample of patients with inflammatory arthritis (IA) and controls. Aims were to examine the link between blood cadmium levels and (1) risk of IA or (2) exacerbation of symptoms and identifying (3) questionnaire and biomarker predictors of IA, and (4) associations of blood cadmium levels with subject characteristics and various biomarkers (oxidative stress, cytokines). This is a lot to accomplish – probably too much, given the relatively small sample size.
The study sample included 51 adult patients with IA who had been hospitalized (reasons for this are not described), and 46 age- and sex-matched controls from other patients from the same hospital (also unclear why they are visiting). Cases were enrolled several years after diagnosis, and there is a lack of information on how the exposure assessment (questionnaires and measured Cd) could be impacted because it is not well described or discussed. One of the main findings is an association of IA with oaffal. Is it possible this food is considered healthy and therefore increased among patients in response to their disease?
As the authors note, it is surprising they do not see the expected association with smoking with IA, which speaks to a possible sampling bias or lack of generalizability in the study sample. Also, it seems odd, since smoking is such an important contributor to blood Cd. What would the association of Cd with IA look like without smokers?
Finally, as the authors note- blood Cd reflects recent exposures more so than urinary levels. While they describe collecting urine, why was this not assessed for Cd as well as a better marker of body burden?
Introduction
Line 51 – “AI” not “IA”?
The background paragraph on cadmium, while informative, is far too long and detailed. Really, the amount of information presented is a distraction – some may belong in the discussion instead (except that the discussion is also too much information to absorb and far too much speculation given the limitations of the study). The authors may wish to consider how to shorten the focus to directly support this particular study. Also, citations that include primary evidence rather than reviews are preferable.
Methods and Results
Were controls inpatients or outpatients? How many patients and controls were contacted to enroll in the study compared to enrolled (i.e., response rate). Can you speak to the representativeness of patients to other types of non-hospitalized patients?
Measured levels are not well described – did all samples have detectable levels? How are non-detects handled in analyses?
What was the source of questions for cadmium exposure assessment? What time period was assessed – lifetime, prior to diagnosis, recent or proximal to diagnosis? The text states “participants declared” environmental exposure – but presumably they simply answered questions about their jobs, etc., not their self-assessed Cd status. Providing questions in a supplemental Table is needed to enable replication of the study methods.
The contribution of Table 2 to the study is not clear – it is expected that cases would differ on various inflammatory parameters.
Table 3 and 4 – are these among patients only or patients and controls?
The multivariable regression model was built using statistical significance rather than a causal framework with confounders. It is unclear why you would want to adjust for possible sources of Cd exposure or outcomes (biomarkers)?
Discussion
As with the introduction, there is simply too much extraneous information here on the potential mechanisms by which Cd could influence IA. The main study design is not strong and so much of this is too much speculation. Clearly there is a lot of information to convey, and perhaps this would work in a separate format such as a review paper.
A note on interpretation: due to the cross-sectional study design and retrospective exposure assessment with indeterminant timing, this study cannot determine risk factors. For rare outcomes, odds ratios can approximate a relative risk, however this conclusion extends beyond the scope of the current work. You cannot say, as in line 324: “the risk of the disease onset decreased…” Instead: “Results showing an increased odds of (outcome) among those with (exposure), suggests that (exposure) may increase risk of (outcome).”
Author Response
Response to Reviewer
I would like to thank the Reviewer for the insightful and helpful review. It has made it possible to correct errors/mistakes and to improve the manuscript. The responses to the Reviewer’s comments are provided below.
This is cross-sectional study of a small convenience sample of patients with inflammatory arthritis (IA) and controls. Aims were to examine the link between blood cadmium levels and (1) risk of IA or (2) exacerbation of symptoms and identifying (3) questionnaire and biomarker predictors of IA, and (4) associations of blood cadmium levels with subject characteristics and various biomarkers (oxidative stress, cytokines). This is a lot to accomplish – probably too much, given the relatively small sample size.
Response:
Thank you for these comments; indeed, the objectives of the study were drawn broadly, as the Reviewer suggested, perhaps too broadly in relation to the size of the study population. We agree with this and we have mentioned in the discussion that the relatively small patient group is one of the limitations of our study. Nevertheless, it was a pilot study, aimed at a preliminary recognition of which of the above-mentioned (investigated) relationships existed and were strong enough to manifest themselves even in a small sample. We realize that stratification analysis and detection of weaker relationships are only possible when investigating a sufficiently large group of subjects.
The study sample included 51 adult patients with IA who had been hospitalized (reasons for this are not described), and 46 age- and sex-matched controls from other patients from the same hospital (also unclear why they are visiting). Cases were enrolled several years after diagnosis, and there is a lack of information on how the exposure assessment (questionnaires and measured Cd) could be impacted because it is not well described or discussed.
Response:
Thank you for this comment. Indeed, neither the study population nor the questionnaire (the questions asked to the patients) have been described in detail. This has already been corrected in the manuscript. The study group consisted of patients in a one-day rheumatology ward who were resistant to routine treatment (nonsteroidal anti-inflammatory drugs, glucocorticoids) and who had been previously or were currently receiving biological therapy or had just been qualified for such treatment. The control group involved patients hospitalized in the internal medicine department (diagnosed in the sleep laboratory) with the aim of verifying the suspicion of obstructive sleep apnoea. They stayed for about a day in the ward for the polysomnographic study. The questionnaire involved questions concerning potential present or past (within the previous 10 years) sources of cadmium exposure (although probably not all possibilities have been covered). Thank you for pointing out that the reader, without seeing the questionnaire, does not receive full information on the potential exposure of the patients to cadmium. The questionnaire has already been translated and submitted as supplementary material.
One of the main findings is an association of IA with oaffal. Is it possible this food is considered healthy and therefore increased among patients in response to their disease?
Response:
Thank you for this comment. We searched for differences in cadmium exposure between the control group and the cases. The more frequent consumption of offal (a potential source of cadmium) by the IA patients than by the control group was the only significant difference that we observed. In the general opinion, eating liver is healthy for individuals with anaemia and iron deficiency. Nutrition specialists point to liver as a source of numerous vitamins and minerals. In the case of fish, nutritionists conclude that the benefits of eating it outweigh any possible harm resulting from metal contamination. I have not encountered such a statement in relation to liver. It is, however, most likely that individuals with chronic diseases (including autoimmune diseases), which are often accompanied by anaemia and low iron levels, consume liver to improve blood parameters. This comment has already been included in the text.
As the authors note, it is surprising they do not see the expected association with smoking with IA, which speaks to a possible sampling bias or lack of generalizability in the study sample. Also, it seems odd, since smoking is such an important contributor to blood Cd.
Response:
Thank you for highlighting the authors’ overly simple and inappropriate conclusion that higher Cd-B levels in IA patients cannot be explained by smoking. Although we asked the respondents about the number of cigarettes smoked and years of smoking, the number of smokers and former smokers in both groups was too small to allow an analysis of the effect of these important variables on blood cadmium levels. Indeed, without this analysis, by comparing only the number of smokers among the cases and in the control group, one cannot draw conclusions about the relationship between smoking and blood cadmium levels, especially as a link between smoking and the occurrence and severity of autoimmune diseases (including joint diseases) has been observed and described in the literature. The inappropriate conclusion has already been removed from the text.
What would the association of Cd with IA look like without smokers?
Response:
The question that the Reviewer asked is interesting and certainly worth investigating. Without individuals with high blood cadmium levels (smokers), a more balanced cadmium level might be expected among patients with IA; documenting such a relationship would require a large group of patients.
Finally, as the authors note- blood Cd reflects recent exposures more so than urinary levels. While they describe collecting urine, why was this not assessed for Cd as well as a better marker of body burden?
Response:
Thank you for this comment. We determined cadmium in urine, but we could not use these results because they were not reliable. In accordance with recommendations, urinary cadmium should be converted to grams of creatinine. For reasons beyond our control, however, urine creatinine was not determined in the central laboratory and we did not have these results available. Nevertheless, it seems that urinary cadmium concentration is not necessarily and not always a better marker of the body burden of this metal. It mainly reflects past exposure, whereas blood cadmium levels point at both past and current exposure. I certainly agree with the Reviewer that it would be good to be able to use both values in the analyses (as we planned). The collected urine was also used to determine 8-hydroxy-2'-deoxyguanosine (8-OHdG).
Introduction
Line 51 – “AI” not “IA”?
Response:
Thank you for noticing this mistake. AI has been corrected to IA.
The background paragraph on cadmium, while informative, is far too long and detailed. Really, the amount of information presented is a distraction – some may belong in the discussion instead (except that the discussion is also too much information to absorb and far too much speculation given the limitations of the study). The authors may wish to consider how to shorten the focus to directly support this particular study. Also, citations that include primary evidence rather than reviews are preferable
Response:
Thank you for this comment. The introduction and information on cadmium have been shortened. The introduction contains literature references to both original papers and reviews. Reviews were cited to present certain concepts, mechanisms, which, in this case, is more conclusive than depicting them on the basis of original but isolated experiments. We agree with the Reviewer that the discussion is long and full of diverse considerations. We may have taken too literally the following guidelines for authors:
‘Discussion: Authors should discuss the results and how they can be interpreted in perspective of previous studies and of the working hypotheses. The findings and their implications should be discussed in the broadest context possible and limitations of the work highlighted. Future research directions may also be mentioned’
Following the Reviewer’s recommendation, the discussion has been partly shortened.
Methods and Results
Were controls inpatients or outpatients? How many patients and controls were contacted to enroll in the study compared to enrolled (i.e., response rate). Can you speak to the representativeness of patients to other types of non-hospitalized patients?
Measured levels are not well described – did all samples have detectable levels? How are non-detects handled in analyses?
What was the source of questions for cadmium exposure assessment? What time period was assessed – lifetime, prior to diagnosis, recent or proximal to diagnosis? The text states “participants declared” environmental exposure – but presumably they simply answered questions about their jobs, etc., not their self-assessed Cd status. Providing questions in a supplemental Table is needed to enable replication of the study methods.
The contribution of Table 2 to the study is not clear – it is expected that cases would differ on various inflammatory parameters.
Table 3 and 4 – are these among patients only or patients and controls?
The multivariable regression model was built using statistical significance rather than a causal framework with confounders. It is unclear why you would want to adjust for possible sources of Cd exposure or outcomes (biomarkers)?
Response:
Thank you for these comments. As indicated previously, the study group consisted of patients in a one-day rheumatology ward who were resistant to routine treatment (nonsteroidal anti-inflammatory drugs, glucocorticoids) and who had been previously or were currently receiving biological therapy or had just been qualified for such treatment. The control group involved patients hospitalized in the internal medicine department (diagnosed in the sleep laboratory) with the aim of verifying the suspicion of obstructive sleep apnoea. They stayed for about a day in the ward for the polysomnographic study. Both the patients and the control subjects were recruited for the study immediately before admission to the ward. Only 3 of the people asked refused to participate and to complete the questionnaire. Patients with IA were more severe cases; as mentioned, they were resistant to routine treatment, treated with or qualified for biologic therapy. The group was representative of such patients, although not of all patients with IA.
The questionnaire involved questions concerning potential present or past (within the previous 10 years) sources of cadmium exposure (although probably not all possibilities have been covered). The questionnaire has already been translated and submitted as supplementary material.
Only for a few samples from the control group were there difficulties in reading the cadmium level from the curve. Samples below the detection threshold were treated as 0.
In Table 2, the level of significance of differences in the values of biochemical parameters (P), including inflammatory markers, is shown in column 4. Most of these differences between the cases and controls, especially for inflammatory markers, were statistically significant (P < 0.05). I am sorry, I thought that the table was quite clearly arranged.
Tables 3 and 4 refer to the whole population. This has already been corrected in the text.
The confounding factors (age, gender) were attempted to be eliminated as far as possible by selecting the cases and the control group. By means of logistic regression, the authors wanted to identify some characteristics of an IA patient (as the Reviewer suggested with the questions, the one refractory to conventional treatment; we intended to determine as many characteristics as the sample size allowed). The predictors taken into account were related to IA (although not in an obvious way, e.g. those constituting diagnostic criteria were not considered), and care was taken that they were not strongly related to each other. Finally, those that met these criteria and statistically significantly correlated with IA were selected to build the model.
Discussion
As with the introduction, there is simply too much extraneous information here on the potential mechanisms by which Cd could influence IA. The main study design is not strong and so much of this is too much speculation. Clearly there is a lot of information to convey, and perhaps this would work in a separate format such as a review paper.
Response:
Thank you for this remark. The discussion has been somewhat shortened. As mentioned earlier, we were guided by the instructions for authors, which recommend that in the discussion, the results and their implications should be presented in the widest possible context.
A note on interpretation: due to the cross-sectional study design and retrospective exposure assessment with indeterminant timing, this study cannot determine risk factors. For rare outcomes, odds ratios can approximate a relative risk, however this conclusion extends beyond the scope of the current work. You cannot say, as in line 324: “the risk of the disease onset decreased…” Instead: “Results showing an increased odds of (outcome) among those with (exposure), suggests that (exposure) may increase risk of (outcome).”
Response:
Thank you for this comment. We agree with the Reviewer on the inaccuracy of the quoted statement. Claims such as that in line 324 have been corrected as suggested by the Reviewer.

Reviewer 2 Report
The manuscript is a publication and analysis of correlations between cadmium exposure and correlation with Inflammatory Arthritis and includes analysis of inflammatory biomarker levels associated with inflammatory pathology. It is well written. I have minor suggestions for revision.
- Discussion, page 6, lines 413-415: The authors state that (though B-Cd concentrations were higher in IA patients than in controls, frequency of smoking was not significantly different from control group. They then conclude that the higher levels of B-Cd among IA patients cannot be attributed to smoking, although they state in the Introduction, lines 61-64, that there is a well documented link between cigarette smoking and arthritis, and between environmental cadmium exposure and arthritis. Perhaps this was due to a slightly flawed approach in this assessment. Other variables need to be considered before arriving at this conclusion, since, as the authors state, smoking is the greatest source of environmental exposure to cadmium to non-industrially exposed populations. There were smokers in both the control and IA patient groups. Paschal et al., Exposure of the U.S. Population Aged 6 years and older to cadmium: 1988-1994, 2000, Arch. Environ. Contam. Toxicol. 38: 377-383, DOI: 10.1007/s002449910050 showed that urine cadmium concentrations increase with age due to cadmium's long biological half life whether or not a person smoke. However, cadmium accumulates at a faster rate among smokers, who have a higher exposure. Although urine cadmium concentration is not the same as serum cadmium concentration, the point is that the body burden of cadmium increases with exposure. Levels of chronic exposure to cadmium are not dependent on smoking status alone, but age, how many years one has been a smoker, and how many cigarettes a day are smoked (this can be approximately measured in pack years). So a smoker who smokes 5 cigarettes a day for one month is considered a smoker, but that smoker will not receive the same cadmium exposure from smoking as a person who smokes 20 cigarettes a day for 20 years. Therefore, the authors are not justified in stating that the lack of difference between prevalence of smoking among IA patients and control subjects suggested that elevated "blood" (actually serum) cadmium among IA patients could not be explained by smoking. In order to be justified in making the conclusion that elevated serum cadmium among IA patients was not due to smoking, the authors would have to include pack years smoked by the individuals with and without IA. They should either provide this data along with statistical analysis, or remove the unsupported conclusion in lines 413-416. Since there was a higher percentage of IA patients who were nonsmokers, it could be stated that among nonsmokers, consumption of offal might be a major source of cadmium exposure, especially among those who consume kidney, since the kidney is the major target organ for cadmium and cadmium bioaccumulates in kidney. To illustrate this, even among smokers, cadmium bioaccumulates in kidneys at 30-60 times the concentrations in lungs, the primary organ of exposure.
- Discussion, page 9, lines 570-580: The authors have the right idea, but have expressed the associations poorly. The statements sound like effects of cadmium exposure or inflammation are causes of increases or decreases in risk for IA, when in fact, they are associated factors. Cd-B is an exposure. If MCH is decreased by Cd-B exposure, and IA is increased by higher Cd-B concentrations, then they are associated effects. Lower Cd-B would decrease the risk of IA, but an increase in MCH did not necessarily reduce the risk of IA. Increased MCH and reduced IA are both consequences of lower Cd-B. So instead of saying that higher MCH, or lower 8-OHdG increase the IA risk, it should be stated that they are "associated with" lower IA risk. This is more consistent with the authors explanation in lines 581-586 that 8-OHdG is a biomarker (a marker of effect) of damage from exposure. In line 590, the term "positively correlated is an example of a better way to express these relationships.
Author Response
Dear Reviewer,
thank you very much for your valuable comments, which have helped eliminate errors and inaccuracies and improve the manuscript.
The manuscript is a publication and analysis of correlations between cadmium exposure and correlation with Inflammatory Arthritis and includes analysis of inflammatory biomarker levels associated with inflammatory pathology. It is well written. I have minor suggestions for revision.
Discussion, page 6, lines 413-415: The authors state that (though B-Cd concentrations were higher in IA patients than in controls, frequency of smoking was not significantly different from control group. They then conclude that the higher levels of B-Cd among IA patients cannot be attributed to smoking, although they state in the Introduction, lines 61-64, that there is a well documented link between cigarette smoking and arthritis, and between environmental cadmium exposure and arthritis. Perhaps this was due to a slightly flawed approach in this assessment. Other variables need to be considered before arriving at this conclusion, since, as the authors state, smoking is the greatest source of environmental exposure to cadmium to non-industrially exposed populations. There were smokers in both the control and IA patient groups. Paschal et al., Exposure of the U.S. Population Aged 6 years and older to cadmium: 1988-1994, 2000, Arch. Environ. Contam. Toxicol. 38: 377-383, DOI: 10.1007/s002449910050 showed that urine cadmium concentrations increase with age due to cadmium's long biological half life whether or not a person smoke. However, cadmium accumulates at a faster rate among smokers, who have a higher exposure. Although urine cadmium concentration is not the same as serum cadmium concentration, the point is that the body burden of cadmium increases with exposure. Levels of chronic exposure to cadmium are not dependent on smoking status alone, but age, how many years one has been a smoker, and how many cigarettes a day are smoked (this can be approximately measured in pack years). So a smoker who smokes 5 cigarettes a day for one month is considered a smoker, but that smoker will not receive the same cadmium exposure from smoking as a person who smokes 20 cigarettes a day for 20 years. Therefore, the authors are not justified in stating that the lack of difference between prevalence of smoking among IA patients and control subjects suggested that elevated "blood" (actually serum) cadmium among IA patients could not be explained by smoking. In order to be justified in making the conclusion that elevated serum cadmium among IA patients was not due to smoking, the authors would have to include pack years smoked by the individuals with and without IA. They should either provide this data along with statistical analysis, or remove the unsupported conclusion in lines 413-416. Since there was a higher percentage of IA patients who were nonsmokers, it could be stated that among nonsmokers, consumption of offal might be a major source of cadmium exposure, especially among those who consume kidney, since the kidney is the major target organ for cadmium and cadmium bioaccumulates in kidney. To illustrate this, even among smokers, cadmium bioaccumulates in kidneys at 30-60 times the concentrations in lungs, the primary organ of exposure.
Response:
Thank you for highlighting the authors’ overly simple and inappropriate conclusion that higher Cd-B levels in IA patients cannot be explained by smoking. Although we asked the respondents about the number of cigarettes smoked and years of smoking, the number of smokers and former smokers in both groups was too small to allow an analysis of the effect of these important variables on blood cadmium levels. Indeed, without this analysis (as the Reviewer rightly pointed out), by comparing only the number of smokers among the cases and in the control group, one cannot draw conclusions about the relationship between smoking and blood cadmium levels, especially as a link between smoking and the occurrence and severity of autoimmune diseases (including joint diseases) has been observed and described in the literature. The inappropriate conclusion has already been removed from the text.
Discussion, page 9, lines 570-580: The authors have the right idea, but have expressed the associations poorly. The statements sound like effects of cadmium exposure or inflammation are causes of increases or decreases in risk for IA, when in fact, they are associated factors. Cd-B is an exposure. If MCH is decreased by Cd-B exposure, and IA is increased by higher Cd-B concentrations, then they are associated effects. Lower Cd-B would decrease the risk of IA, but an increase in MCH did not necessarily reduce the risk of IA. Increased MCH and reduced IA are both consequences of lower Cd-B. So instead of saying that higher MCH, or lower 8-OHdG increase the IA risk, it should be stated that they are "associated with" lower IA risk. This is more consistent with the authors explanation in lines 581-586 that 8-OHdG is a biomarker (a marker of effect) of damage from exposure. In line 590, the term "positively correlated is an example of a better way to express these relationships.
Response:
Thank you for pointing out this illogical wording. This was probably due to a hurried and too simple interpretation of the equation obtained. The inappropriate phrasing has already been corrected as suggested by the Reviewer.

Reviewer 3 Report
“Cadmium body burden and inflammatory arthritis: a pilot study in patients from Lower Silesia, Poland” by Markiewicz-Górka et al. highlights the mechanisms of cadmium effect on the development of autoimmune joint diseases. The article is very well written and very interesting.
I have only some minor remarks:
- Lines 50-51: “Genetic factors play an essential role in autoimmunity but they 50 do not explain all the differences in AI prevalence among populations [9]” – IA instated of AI.
- Line 80: When you have more than two references, you can write [22-25], instated of [22,23,24,25]. The same for line 83.
- You should add the novelty of the article. What had you discovered new compared with the mentioned studies?
- I think you should rearrange equation (2).
- Being a long article, I think is useful to add an abbreviations section.
Author Response
Dear Reviewer,
thank you for your positive review, comments, and suggestions. We have improved our paper in accordance with the remarks.
Cadmium body burden and inflammatory arthritis: a pilot study in patients from Lower Silesia, Poland” by Markiewicz-Górka et al. highlights the mechanisms of cadmium effect on the development of autoimmune joint diseases. The article is very well written and very interesting.
I have only some minor remarks:
Lines 50-51: “Genetic factors play an essential role in autoimmunity but they 50 do not explain all the differences in AI prevalence among populations [9]” – IA instated of AI.
Response:
Thank you for highlighting this mistake. It has already been corrected in the manuscript.
Line 80: When you have more than two references, you can write [22-25], instated of [22,23,24,25]. The same for line 83.
Response:
Thank you for this comment. The notation of literature references has been changed in the text as suggested by the Reviewer.
You should add the novelty of the article. What had you discovered new compared with the mentioned studies?
Response:
Thank you for this comment. A new observation in our study appears to be the demonstration of a difference in the effect of high blood cadmium concentrations on anti-inflammatory IL-10 levels in the IA patients and controls. High Cd-B (≥ upper quartile) significantly reduced IL-10 levels in the IA patients and had no effect on IL-10 or even increased its levels (not statistically significantly) in the control group. This phenomenon may be related to a higher sensitivity to cadmium in individuals who are predisposed to the development of autoimmune joint diseases.
I think you should rearrange equation (2).
Response:
Thank you for this comment. The mathematical notation of the equation has been submitted in the supplementary material. The notation in the manuscript follows the guidelines for authors, which instruct them to write equations in a sequence, in one line.
Being a long article, I think is useful to add an abbreviations section.
Response:
Thank you for this comment. The abbreviations section has been appended to the manuscript.
